# Risk factors for emerging intraocular inflammation after intravitreal brolucizumab injection for age-related macular degeneration

**Ryo Mukai** *, **Hidetaka Matsumoto** , **Hideo Akiyama**

Department of Ophthalmology, Gunma University Graduate School of Medicine, Maebashi, Gunma, Japan

* rmukai@gunma-u.ac.jp

## Abstract

### Purpose

To analyze the risk factors associated with emerging intraocular inflammation (IOI) after intravitreal brolucizumab injection (IVBr) to treat age-related macular degeneration (AMD).

### Methods

This study included 93 eyes of 90 patients. The incidence of emerging IOI was analyzed. The patients were classified into IOI or non-IOI groups, and background clinical characteristics in each group were compared.

### Results

IOI occurred in 14 eyes of 14 cases (16%; five women, nine men [5:9]; IOI group) after IVBr; contrastingly, no IOI occurred in 76 patients (10 women, 66 men [10:66]; non-IOI group). The mean ages in IOI and non-IOI groups were 79.4 ± 8.1 and 73.8 ± 8.9 years old, respectively, and the average age in the IOI group was significantly higher than that in the non-IOI group (P = 0.0425). In addition, the percentages of females in the IOI and non-IOI groups were 43% and 13%, respectively, and IOI occurred predominantly in females (odds ratio: 4.95, P = 0.0076). Moreover, the prevalence of diabetes in the IOI and non-IOI groups was 64% and 32%, respectively, with a significant difference (odds ratio: 3.90, P = 0.0196). In contrast, the prevalence of hypertension in the IOI and non-IOI groups was 36% and 57%, respectively, with no significant difference (P = 0.15).

### Conclusion

The comparison of clinical profiles of IOI or non-IOI cases in IVBr treatment for AMD suggests that the risk factors for IOI are old age, female sex, and history of diabetes; however, IOI with vasculitis or vascular occlusion in this cohort does not seem to cause severe visual impairment. Further studies are required to investigate potential risk factors for IOI.

**Data Availability Statement:** All relevant data are within the manuscript and its Supporting Information files

**Funding:** The authors received no specific funding for this work.

**Competing interests:** The authors have declared that no competing interests exist.

## Introduction

Age-related macular degeneration (AMD) is currently the most prevalent cause of blindness in both the western and eastern hemispheres [1, 2]. Since 2006, intravitreal injection of anti-vascular endothelial growth factor (VEGF) antibodies is the major treatment for patients with neovascular AMD [3, 4]. In this era, visual acuity can improve in most patients after continuous treatment with anti-VEGF agents for several years, and this improvement can be maintained for at least 7 years [5–7]. However, treatment with anti-VEGF drugs such as ranibizumab and aflibercept can potentially induce vascular occlusive diseases and severe adverse events including cardiovascular events or cerebral infarction; although the frequencies of these events are very low [8–10].

Recently, brolucizumab, a novel anti-VEGF agent, has been shown to have a prolonged effect and is potentially effective against choroidal neovascularization beneath the retinal pigment epithelium, as shown in phase 3 studies such as HAWK and HARRIER [11]. However, in these reports, unexpected cases of intraocular inflammation (IOI) after intravitreal brolucizumab (IVBr) injection have been noted having an incidence of 4.6% [12]; this was higher than that of ranibizumab and aflibercept injection groups (1.5% and 0.5–1.1% respectively) [13]. To date, the clinical profiles of these cases are unclear.

This report clarifies the risk factors for emerging IOI after IVBr injection.

## Methods

This retrospective study obtained institutional review board approval from the Gunma University Graduate School of Medicine and adhered to the Declaration of Helsinki. The requirement for written informed consent was waived owing to the retrospective nature of the study. All patients with a clinical diagnosis of typical AMD, polypoidal choroidal neovasculopathy (PCV), and pachychoroid neovasculopathy (PNV) at the Department of Ophthalmology of Gunma University Medical Hospital between June 2020 and January 2021 were included in this study. All participants were examined using a fundus ophthalmoscope, fluorescein angiography (FA), indocyaninegreen angiography (IA; Heidelberg Engineering, Heidelberg, Germany), and swept-source optical coherence tomography (SS-OCT; Plex Elite 9000; Carl Zeiss Meditec, Dublin, CA, USA) incorporating a tunable laser with a central wavelength of 1050 nm and acquiring 100,000 A-scans/s. SS-OCT has an axial resolution of 1.95 μm and a lateral resolution of 20 μm. SS-OCT volume images were obtained using a raster scan protocol of $500 \times 500$ B scans per second, covering an area of $6 \times 6$ mm centered on the fovea.

Three monthly injections of brolucizumab (Beovu, 6.0 mg/0.05 mL; Novartis, Basel, Switzerland) were administered as a loading phase treatment for treatment-naïve patients. After three months injections, patients were followed-up using Treat and Extend (TAE) regimen. If persistent signs of active disease were present, injections were continued bimonthly until the macula was dry. Once the macula was dry, the interval between injections increased to 4 weeks initially and the maximum interval was 16 weeks. Contrastingly, for non-treatment naïve cases, TAE regimen was applied after first brolucizumab injection.

In this study, 7 physicians performed IVBr, in accordance with guidelines for intravitreal injection for macular disease [14].

Patients were monitored for emergence of IOI starting from one week after the first injection and monthly thereafter for 3 months. For the purpose of our study, IOI included iritis, vitritis, and vasculitis or occlusion of vessels. To precisely detect IOI, a detailed slit lamp test and fundus examination were performed. Additionally, we analyzed the patients' eyes with an ultra-wide field scanning laser ophthalmoscope (Optos 200Tx; Optos, Dunfermline, Scotland), during monitoring, to broadly detect vascular changes. To diagnose occlusion of vessels as

IOI, we performed FA and IA angiography at the initial visit and at the visit after the loading dose. SS-OCT (DRI-OCT triton, Tokyo, Japan) were performed at every visit.

To determine whether patients had diabetes (HbA1c) or hypertension, we assessed medical records in all cases. In addition, before performing the first angiography, systolic and diastolic blood pressures were checked in all cases.

## Statistics

The Mann-Whitney U test was used to compare the mean age between the IOI and non-IOI groups. A chi-square test was performed to analyze the odds ratio and p-value for the prevalence of diabetes and hypertension and the dominance of males or females in each group. Data analysis was performed using GraphPad Prism version 9 software (GraphPad Software, La Jolla, CA, USA).

## Results

Ninety-three eyes of 90 patients were included in this study. Seventy five eyes (81%) were treatment naïve. IOI occurred in 14 eyes of 14 cases (16%; five women, nine men [5:9]; IOI group) after IVBr; contrastingly, IOI did not occur in 76 cases (10 women, 66 men [10:66]; non-IOI group) during 3 months. The mean ages in IOI and non-IOI groups were 79.4 ± 8.1 and 73.8 ± 8.9 years, respectively, and the average age in the IOI group was significantly higher (P = 0.0425) than that in non-IOI group. In addition, the percentages of females in the IOI and non-IOI groups were 43% and 13%, respectively, and IOI occurred predominantly in females (odds ratio: 4.95, P = 0.0076). Moreover, the prevalence of diabetes in the IOI and non-IOI groups was 64% and 32%, respectively, with a significant difference (odds ratio: 3.90, P = 0.0196). In contrast, the prevalence of hypertension in the IOI and non-IOI groups was 36% and 57%, respectively, with no significant difference (P = 0.15). The mean systolic and diastolic blood pressures before the first injection in the IOI and non-IOI groups were 133 ± 18/74 ± 11 mm of Hg and 141 ± 17/79 ± 12 mm of Hg, respectively, without a significant difference (P = 0.11 and 0.10, respectively) (Tables 1 and 2).

**Table 1. Comparison of clinical characteristics between cases with and without intraocular inflammation after brolucizumab injection.**

|  | Total | With IOI | Without IOI | P value |
|---|---|---|---|---|
| **Patients** | 90 | 14 | 76 | |
| **Number(eyes)** | 93 | 14 | 79 | |
| **Age** | 74.8±9.0 | 79.4±8.1 | 73.8±8.9 | 0.0425 |
| **Female: Male** | 15:75 | 5:9 | 10:66 | 0.0076 |
| **General condition** | | | | |
| **Diabetes** | 32(36%) | 9(64%) | 23(32%) | 0.0196 |
| **Hypertension** | 45(50%) | 5(36%) | 40(57%) | 0.15 |
| **Systolic pressure** | 139±17 | 133±18 | 141±17 | 0.11 |
| **Diastolic pressure** | 78±12 | 74±11 | 79±12 | 0.1 |
| **Lesion types** | | | | 0.74 |
| **Typical AMD** | 32 | 4 | 28 | |
| **PCV** | 44 | 7 | 37 | |
| **PNV** | 14 | 3 | 11 | |

IOI: intraocular inflammation. PCV: polypoidal choroidal vasculopathy, PNV: pachychoroid neovasculopathy.

**Table 2. Clinical profiles of cases with intraocular inflammation after brolucizumab injection.**

| General conditions | Liver dysfunction | DM (Hba1c8.9) | Post CI('98), HT,DM (HbA1c:7.1%) | HT,DM (HbA1c:7.2%) | Border line DM (HbA1c:5.7%) | Alzheimer (HbA1c:6.4%) | Liver dysfunction, Thrombocytopenia | Border line DM (BS;175, HbA1c:5.5%) | HT, CKD | DM (HbA1c:7.2%), HT,HL,HU | DM (HbA1c:8.3%) | DM(HbA1c:7.7%), Post coronary artery bypass ICA stenosis, Old CI('10),RA | Old MCI ('18), ASO | DM (HbA1c:10.7), HT,HU |
|---|---|---|---|---|---|---|---|---|---|---|---|---|---|---|
| BCVA at the last visit | 20/25 | 20/2000 | 12.5/20 | 20/40 | 25/20 | 25/20 | 4/6.3 | 20/25 | 20/25 | 20/25 | 4/6.3 | 16/20 | 20/40 | 20/63 |
| BCVA at the initial visit | 20/63 | 20/1000 | 20/40 | 20/50 | 4/6.3 | 20/20 | 20/25 | 20/20 | 4/6.3 | 4/6.3 | 20/32 | 4/6.3 | 20/50 | 20/63 |
| Local steroid | + | 0 | + | + | + | + | 0 | + | + | + | + | + | + | + |
| Vascular occlusion | 0 | 0 | + | 0 | + | + | 0 | 0 | 0 | 0 | 0 | 0 | 0 | 0 |
| Periphlebitis | + | 0 | + | 0 | + | + | 0 | + | 0 | 0 | 0 | 0 | + | 0 |
| Arteritis | + | + | + | 0 | + | + | 0 | + | + | 0 | + | 0 | + | + |
| Vitritis | + | + | + | 0 | + | + | 0 | + | + | + | + | + | + | + |
| Iritis | 0 | 0 | 0 | + | + | + | + | + | 0 | + | 0 | 0 | + | + |
| IOI onset (days) after last injection | 20 | 27 | 28 | 5 | 28 | 26 | 12 | 21 | 19 | 25 | 28 | 15 | 30 | 3 |
| Number of Injections prior to IOI | 2 | 1 | 2 | 1 | 3 | 3 | 1 | 1 | 1 | 1 | 2 | 2 | 1 | 3 |
| Lesion | PCV | Occult | PCV | PCV | PCV | PCV | Occult | PNV | PCV | PNV | PNV | Occult | Occult | PCV |
| Naïve/Switch | Naïve | Naïve | Naïve | Naïve | Naïve | Naïve | Switch | Switch | Naïve | Naïve | Naïve | Naïve | Naïve | Naïve |
| Sex | F | M | M | M | F | F | M | F | M | M | F | M | F | M |
| Age | 77 | 86 | 88 | 76 | 72 | 90 | 65 | 72 | 94 | 75 | 78 | 77 | 76 | 85 |
| Cases | 1 | 2 | 3 | 4 | 5 | 6 | 7 | 8 | 9 | 10 | 11 | 12 | 13 | 14 |

IOI: intraocular inflammation, PCV: polypoidal choroidal vasculopathy, BCVA: best corrected visual acuity, DM: diabetes mellitus, CI: cerebral infarction, HT: hypertension, CKD: chronic kidney disease, HL: hyperlipidemia, HU: hyperuremia, ICA: internal carotid artery, RA: rheumatoid arthritis, ASO: arteriosclerosis obliterans.

## Discussion

Overall, IOI occurred in 16% of AMD cases treated with IVBr injections. Female sex, history of diabetes, and older age were risk factors for emerging IOI after IVBr in our cohort.

In IOI cases, blurred vision, eye floaters, ocular pain, and conjunctival injection were reported as the primary symptoms. Particularly, 11/14 patients (79%) claimed to experience eye floaters, but no patients complained of pain. Neither hypopyon nor ciliary injection was detected in these patients. Based on the aforementioned findings, it is unlikely that these patients had endophthalmitis.

In the Beovu safety site [15], IOI was reported in 4.3% of cases of total injections (HAWKS and HARRIER) [12]. At this site, 5.1/10000 shots were accompanied by vasculitis and 3.4/10000 with occlusive vasculitis. In our previous report, IOI occurred in 19% of cases (8/42 eyes), primarily in women [16]. Furthermore, in an early experience study, IOI was observed in 8.1% of cases (14/172 eyes), and the female sex showed a relative risk of 1.27 for IOI [17]. Moreover, the American Society of Retina Specialists analyzed reports of inflammation following IVBr injection for neovascular AMD [18]. In this analysis, 26 eyes showed IOI, and 22 eyes with IOI occurred in women. Tendency and female dominance were similar to those reported previously. However, Maruko reported that IOI was detected in 9% (4/43 eyes) of treatment-naïve patients and 10% in a switched group, and in his report, no sex-related disease association was observed [19].

IOI with vasculitis or vascular occlusion in this cohort did not seem to cause severe visual impairment (Table 2). Additionally, IOI with vasculitis did not worsen the visual acuity. Early treatment with steroids could relieve IOI and restore visual outcome. In two cases with reduced vision after IOI, exudative macular neovascularization seemed to contribute to the visual deterioration. Old age, female sex, and history of diabetes seemed to contribute to the occurrence of IOI in this cohort; however, these factors did not seem to be associated with vision threatening in the eyes with IOI.

During treatment with protein therapeutics, the emergence of anti-drug antibodies (ADA) has been the focus of recent attention because ADA potentially works as a neutral antibody for a parent drug or an inducer of adverse reactions [20]. With the use of ADA against tumor necrosis factor-α (TNF-α) inhibitors for treatment of rheumatoid arthritis, TNF-α was detected in a maximum of 87% of cases [21]. In the form submitted to the FDA where brolucizumab was filed as a new drug, it was suggested that ADA may form after the use of brolucizumab [22]. Sharma speculated that ADA might be associated with IOI in the use of brolucizumab [23].

Generally, there may be sex differences in the immune response. Taken together, males and females may differ in their responses to brolucizumab injection.

Higher age can be a risk factor for emerging IOIs in this cohort. The mean ages of the cohorts used in the HAWK and HARRIER studies [12] were 76.6 and 74.8 years, respectively, and the average age of IOI cases reported by *Ophthalmology* was higher than these ages [12]. In four vision loss cases, after treatment with brolucizumab, the ages were 88, 92, 77, and 76 years [22–25]. Higher age has been identified as a high-risk factor for cardiovascular events [26]. In addition, one of the positive predictors of stroke after myocardial infarction is advanced age [20]. Overall, higher age can potentially be a high-risk factor, not only for cardiovascular events, but also for occlusive events, even in the retina and choroid.

Diabetes can accompany microangiopathy and macroangiopathy, which are caused by sclerosis [27, 28]. Hyperglycemia due to diabetes, tissue resistance against insulin, hyperinsulinemia accompanied with insulin tolerance, and obesity, specifically visceral fat obesity, can promote arteriosclerosis in diabetes [29]. Mortality due to coronary artery diseases and the

frequency of cardiac infarction increase in cases of diabetes [30]. Advanced glycation end-products or glycation proteins can induce the release of pathophysiological substances such as IL-6, IL-8, MCP-1, ICAM-1, and IP-10 in conditions such as diabetic retinopathy [31]. In addition, microinflammation can cause diabetic retinopathy mediated by circulating mono-cytes, tissue-resident macrophages, and monocyte-derived inflammatory macrophages [32]. Patients with severe diabetes mellitus sometimes have accompanying iritis clinically, with hyperpermeability due to impairment of blood aqueous humor barrier due to damage of endo-thelial cells in capillary vessels. Histopathological sections of the vitreous, aspirated from patients with vasculitis that developed after brolucizumab injection, revealed infiltration of CD20 positive B cells, CD3, 4, and 8 positive T cells and CD68 positive histiocytes [33]. IOI consists of iritis, OCV, vasculitis, and vascular occlusion, and such pathogeneses if accompa-nied by diabetes mellitus can potentially promote further inflammation. Of the 14 patients with IOI after brolucizumab injection, nine had diabetes mellitus, and diabetic retinopathy progressed in the retina in 6/9 of these cases (67%). Based on these findings, we believe that IVBr injections in patients with diabetes mellitus may result in occlusive or inflammatory changes in the retina.

In conclusion, the risk factors for emerging IOI after IVBr injection are old age, female sex, and history of diabetes; however, IOI with vasculitis or vascular occlusion in this cohort does not seem to cause severe visual impairment. Further studies are required to investigate poten-tial risk factors for IOI. In almost 80% of the cases, patients had eye floaters before or after the onset of the IOI. Thus, it is important to pay attention to the claims of floaters after IVBr injection.

## Supporting information

**S1 Table. Clinical characteristics of treatment naïve and non-treatment naïve cases.**
(PDF)

## Author Contributions

**Conceptualization:** Ryo Mukai, Hidetaka Matsumoto.

**Data curation:** Ryo Mukai, Hidetaka Matsumoto.

**Formal analysis:** Ryo Mukai, Hidetaka Matsumoto.

**Investigation:** Ryo Mukai.

**Supervision:** Hideo Akiyama.

**Writing – original draft:** Ryo Mukai.

**Writing – review & editing:** Hidetaka Matsumoto, Hideo Akiyama.

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
