## [Decision Letter · Decision Letter 0]

14 Jul 2021

PONE-D-21-16802

Risk factors for emerging intraocular inflammation after intravitreal bevacizumab injection for age-related macular degeneration

PLOS ONE

Dear Dr. Mukai,

Thank you for submitting your manuscript to PLOS ONE. After careful consideration, we feel that it has merit but does not fully meet PLOS ONE’s publication criteria as it currently stands. Therefore, we invite you to submit a revised version of the manuscript that addresses the points raised during the review process.

Both reviewers found the topic of great interest but they note many limitations that have to be addressed before a final decision can be made

We look forward to receiving your revised manuscript.

Kind regards,

Demetrios G. Vavvas

Academic Editor

PLOS ONE

Journal Requirements:

2. Thank you for including your ethics statement: "This retrospective study obtained institutional review board approval from the Gunma University Graduate School of Medicine.The data were analyzed anonymously"

a) Please provide additional details regarding participant consent. In the ethics statement in the Methods and online submission information, please ensure that you have specified (1) whether consent was informed and (2) what type you obtained (for instance, written or verbal, and if verbal, how it was documented and witnessed). If your study included minors, state whether you obtained consent from parents or guardians. If the need for consent was waived by the ethics committee, please include this information.

Reviewers' comments:

Reviewer's Responses to Questions

**Comments to the Author**

1. Is the manuscript technically sound, and do the data support the conclusions?

Reviewer #1: Partly

Reviewer #2: No

2. Has the statistical analysis been performed appropriately and rigorously? 

Reviewer #1: I Don't Know

Reviewer #2: I Don't Know

3. Have the authors made all data underlying the findings in their manuscript fully available?

Reviewer #1: No

Reviewer #2: No

4. Is the manuscript presented in an intelligible fashion and written in standard English?

Reviewer #1: Yes

Reviewer #2: Yes

5. Review Comments to the Author

Reviewer #1: The authors present an interesting study on a topic of very high interest: intraocular inflammation after brolucizumab. They are invited to consider the following recommendations to further strengthen their work:

The very title of the article is not in agreement with the article. Do the authors investigate bevacizumab (Avastin) or Brolucizumab (Beovu) ?

Define ’others’ in abstract results

Please correct syntax in the last phrase of your abstract: ‘elders’ is not a ‘risk factor’.

Please be more specific in your introduction tha, ‘relatively frequently’.

Describe the percentages for IOI for bevacizumab, ranibizumab and aflibercept and directly compare with the percentages in the brolucizumab studies.

Seems like not all the eyes included in this study were treatment naive. if not, how many weeks/months since the last antiVEGF injection? define your inclusion and exclusion criteria in more detail.

Was FA, IA and SS-OCTA performed to all eyes at every single visit ? This would be interesting in terms of how many imaging tests were the patients undergoing in their regular visits, since this was a retrospective review, not a prospective study.

Methods should be overall more detailed.

What exactly was the technique/protocol followed when injecting?

Were all the injections preformed by the same person ?

If not, what was the technique/protocol of all different treating physicians exactly the same ?

You should define intraocular inflammation in a very strict and precise manner

It would be interesting to investigate whether the injection technique/protocol had any effect on the occurrence of IOI.

Statistical analysis: did the authors check whether their data are normally distributed ?

The authors need to go more in depth and try to explain why the factors they found that were associated with higher IOI rates are risk factors. Please elaborate more in your discussion.

Reviewer #2: The authors present a study regarding "Risk factors for emerging intraocular inflammation after intravitreal bevacizumab injection for age-related macular degeneration". This is a very interesting topic as we are trying to understand the safety profile of this new agent Brolicizumab even though they have the wrong name of medicine in the title which makes it very confusing...

However, the study the authors have presented has very limited analysis and does not add anything to the literature and is missing significant information eg time when IOI was diagnosed, severity of IOI, number of previous anti VEGF injection, what type of of anti VEGF injection, also majority of oateintas had PCV in the IOI group- was it the same as in the IOI group? What about degreee of existing DR in addition to DM. Authors should do a much more through review of all characteristics of patients to provide an analysis of the risk factors in involved in IOI group.

6. PLOS authors have the option to publish the peer review history of their article (what does this mean?). If published, this will include your full peer review and any attached files.

Reviewer #1: No

Reviewer #2: No

---

## [Author Response · Author response to Decision Letter 0]

25 Aug 2021

Dear Dr. Vavvas,

Thank you for giving me the opportunity to submit a revised draft of my manuscript titled “Risk factors for emerging intraocular inflammation after intravitreal brolucizumab injection for age-related macular degeneration” to PLOS ONE. We appreciate the time and effort that you and the reviewers have dedicated to providing your valuable feedback on my manuscript. We are grateful to the reviewers for their insightful comments on our paper. We have been able to incorporate changes to reflect most of the suggestions provided by the reviewers. We have indicated the changes within the manuscript in red font. Here is a point-by-point response to the reviewers’ comments and concerns.

---

## [Decision Letter · Decision Letter 1]

20 Oct 2021

PONE-D-21-16802R1Risk factors for emerging intraocular inflammation after intravitreal bevacizumab injection for age-related macular degenerationPLOS ONE

Dear Dr. Mukai,

Thank you for submitting your manuscript to PLOS ONE. After careful consideration, we feel that it has merit but does not fully meet PLOS ONE’s publication criteria as it currently stands. Therefore, we invite you to submit a revised version of the manuscript that addresses the points raised during the review process.

 The manuscript has improved. There remain some areas for clarification and expansion and we look forward to hte revised version. 

We look forward to receiving your revised manuscript.

Kind regards,

Demetrios G. Vavvas

Academic Editor

PLOS ONE

Journal Requirements:

Additional Editor Comments (if provided):

Reviewers' comments:

Reviewer's Responses to Questions

**Comments to the Author**

1. If the authors have adequately addressed your comments raised in a previous round of review and you feel that this manuscript is now acceptable for publication, you may indicate that here to bypass the “Comments to the Author” section, enter your conflict of interest statement in the “Confidential to Editor” section, and submit your "Accept" recommendation.

Reviewer #1: (No Response)

Reviewer #2: (No Response)

2. Is the manuscript technically sound, and do the data support the conclusions?

Reviewer #1: Partly

Reviewer #2: Partly

3. Has the statistical analysis been performed appropriately and rigorously? 

Reviewer #1: I Don't Know

Reviewer #2: I Don't Know

4. Have the authors made all data underlying the findings in their manuscript fully available?

Reviewer #1: No

Reviewer #2: No

5. Is the manuscript presented in an intelligible fashion and written in standard English?

Reviewer #1: Yes

Reviewer #2: Yes

6. Review Comments to the Author

Reviewer #1: The authors have substantially improved their work. They are invited to consider the following additional recommendations:

1. Your abstract’s conclusion seems more like a purpose. Please provide a conclusion that sums up the findings of your study and why they might be clinically relevant. (same way as in the conclusion ion the manuscript)

2. Injection technique varies between different treating physicians. This is well documented for antiVEGF injections. How can the authors be sure that injection technique did not affect the rates of IOI since per line 72, 7 different physicians performed the brolucizumab injections.

3. Please clarify the device used for SS-OCT (line 80)

4. Table 2 is confusing. What do all the ‘1’s represent ? If its individual parents please number the patients.

5. Did the number of previous injections in the non-treatment naive eyes affect the rate of IOI ?

Reviewer #2: Even though the manuscript has improved from pervious version, it still lacks in depth analysis in particular regarding the vision outcomes of IOI and the vision threatening forms of IOI of vasculitis and how they responded and if risk factors described for all IOI also apply for the vision threatening one that is the most concerning

7. PLOS authors have the option to publish the peer review history of their article (what does this mean?). If published, this will include your full peer review and any attached files.

Reviewer #1: No

Reviewer #2: No

---

## [Author Response · Author response to Decision Letter 1]

27 Oct 2021

Dear Editor and reviewers,

Reviewer #1: The authors have substantially improved their work. They are invited to consider the following additional recommendations:

1. Your abstract’s conclusion seems more like a purpose. Please provide a conclusion that sums up the findings of your study and why they might be clinically relevant. (same way as in the conclusion ion the manuscript)

Response:

In our cohort, three factors might have influenced the occurrence of IOI after IVBr injections. IOI with vascular occlusion by itself did not seem to cause severe visual impairment in this cohort. Thus, it is likely that these factors were not associated with vision threatening in the eyes with IOI. Considering the possibility of further indication of IVBr for diabetic macular edema, “Further studies are required to investigate potential risk factors for IOI” should be added to the conclusion. We have added this information to the Abstract and Discussion section, as follows

“The comparison of the clinical profiles of IOI or non-IOI cases in IVBr treatment for AMD suggests that the risk factors for IOI are old age, female sex, and history of diabetes; however, IOI with vasculitis or vascular occlusion in this cohort does not seem to cause severe visual impairment. Further studies are required to investigate the potential risk factors for IOI.”

2. Injection technique varies between different treating physicians. This is well documented for antiVEGF injections. How can the authors be sure that injection technique did not affect the rates of IOI since per line 72, 7 different physicians performed the brolucizumab injections.

Response:

As we responded to your comment in the previous revision, we published an article that aimed to standardize the injection techniques for individual physicians. All staff were trained using this manual. We performed 4093 injections of anti VEGF drugs, except for brolucizumab, for one year and no IOIs occurred. We also participated in a multicenter study, which investigated the incidence of endophthalmitis after anti VEGF injections in 2020, but no infections were reported during this period. These results suggested that standardization was successful. 

Ganka Rinsho Kiyo　2018 Vol 11,No 9 P.688 (Domestic journal)

Sci Rep 2020 Dec 17;10 (1): 22122

3. Please clarify the device used for SS-OCT (line 80)

Response:

We used DRI-OCT Triton (Topcon, Tokyo, JAPAN). We have added this information to the Methods section.

4. Table 2 is confusing. What do all the ‘1’s represent ? If its individual parents please number the patients.

Response

We erased “1” and added “+” or “0” instead.

5. Did the number of previous injections in the non-treatment naive eyes affect the rate of IOI ?

Response:

In two patients with IOI in the non-treatment naïve eyes, one month prior to brolucizumab injection, aflibercept was injected. Just before brolucizumab injection, no IOI was detected on slit lamp and fundus examination and optical coherence tomography. IOI in these two cases occurred on 12- and 21-days post brolucizumab injections. Thus, we believe that IOI was mostly associated with brolucizumab injection. 

Reviewer #2: Even though the manuscript has improved from pervious version, it still lacks in depth analysis in particular regarding the vision outcomes of IOI and the vision threatening forms of IOI of vasculitis and how they responded and if risk factors described for all IOI also apply for the vision threatening one that is the most concerning

Response:

Thank you so much for your comment. First, IOI with vascular occlusion in this cohort did not seem to cause severe visual impairment. Additionally, IOI with vasculitis did not worsen the visual acuity. Early treatment with steroids could relieve IOI and restore visual outcome.

In two cases with decreased vision after IOI, exudative change of MNV seemed to contribute to the visual deterioration. Finally, old age, female sex, and history of diabetes seemed to contribute to the occurrence of IOI in this cohort; however, these factors did not seem to be associated with vision threatening in the eyes with IOI. 

We have added this description to the Discussion section.

---

## [Editor Report · Decision Letter 2]

29 Oct 2021

Risk factors for emerging intraocular inflammation after intravitreal brolucizumb injection for age-related macular degeneration

PONE-D-21-16802R2

Dear Dr. Mukai,

We’re pleased to inform you that your manuscript has been judged scientifically suitable for publication and will be formally accepted for publication once it meets all outstanding technical requirements.

Kind regards,

Demetrios G. Vavvas

Academic Editor

PLOS ONE
---

## [Editor Report · Acceptance letter]

9 Nov 2021

PONE-D-21-16802R2 

Risk factors for emerging intraocular inflammation after intravitreal brolucizumab injection for age-related macular degeneration 

Dear Dr. Mukai:

I'm pleased to inform you that your manuscript has been deemed suitable for publication in PLOS ONE. Congratulations! Your manuscript is now with our production department. 

Kind regards, 

on behalf of

Prof. Demetrios G. Vavvas 

Academic Editor

PLOS ONE